# Hierarchical Prototype Networks for Continual Graph Representation Learning

## Abstract

Despite significant advances in graph representation learning, little attention has been paid to graph data in which new categories of nodes (*e.g.*, new research areas in citation networks or new types of products in co-purchasing networks) and their associated edges are continuously emerging. The key challenge is to incorporate the feature and topological information of new nodes in a continuous and effective manner such that performance over existing nodes is uninterrupted. To this end, we present Hierarchical Prototype Networks (HPNs) which can adaptively extract different levels of abstract knowledge in the form of prototypes to represent continually expanded graphs. Specifically, we first leverage a set of Atomic Feature Extractors (AFEs) to generate basic features which can encode both the elemental attribute information and the topological structure of the target node. Next, we develop HPNs by adaptively selecting relevant AFEs and represent each node with three-levels of prototypes, *i.e.*, atomic-level, node-level, and class-level. In this way, whenever a new category of nodes is given, only the relevant AFEs and prototypes at each level will be activated and refined, while others remain uninterrupted. Finally, we provide the theoretical analysis on memory consumption bound and the continual learning capability of HPNs. Extensive empirical studies on eight different public datasets justify that HPNs are memory efficient and can achieve state-of-the-art performance on different continual graph representation learning tasks.

## 1 Introduction

Graph representation learning aims to pursue a meaningful vector representation of each node so as to facilitate downstream applications such as node classification, link prediction, *etc*. Traditional methods are developed based on graph statistics [23] or hand-crafted features [3, 16]. Recently, a great amount of attention has been paid to graph neural networks (GNNs), such as graph convolutional network (GCNs) [12], GraphSAGE [10], Graph Attention Networks (GATs) [31], and their extensions [34, 6, 41, 14, 7, 24, 38]. This is because they can jointly consider the feature and topological information of each node. Most of these approaches, however, focus on static graphs and cannot generalize to the case when new categories of nodes are emerging.

In many real world applications, different categories of nodes and their associated edges (in the form of subgraphs) are often continuously emerging in existing graphs. For instance, in a citation network [27, 32, 20], papers describing new research areas will gradually appear in the citation graph; in a co-purchasing network such as Amazon [4], new types of products will continuously be updated to the graph. Given these facts, how to incorporate the feature and topological information of new nodes in a continuous and effective manner such that performance over existing nodes is uninterrupted is a critical problem to investigate.

To address this issue, various types of continual learning approaches can be considered. Existing continual learning techniques fall into three main categories, *i.e.*, regularization-based methods that penalize (or reward) their model objectives so as to maintain satisfactory performance on previous tasks [11, 9, 26], *e.g.*, Learning without Forgetting (LwF) [15] and Elastic Weight Consolidation (EWC) [13]; memory-replay based methods that constantly feed a model with representative data

Submitted to 35th Conference on Neural Information Processing Systems (NeurIPS 2021). Do not distribute.

or exemplars of previous tasks to prevent them from being forgotten [18, 28, 2, 5, 8], *e.g.*, Gradient Episodic Memory (GEM) [18]; and parametric isolation based methods that adaptively introduce new parameters for new tasks and avoid the existing parameters of previous tasks being drastically changed [25, 36, 35, 33]. Although these approaches exhibited promising performance in mitigating the problem of catastrophic forgetting in different applications, *e.g.*, image classification, action recognition, and reinforcement learning, they are not suitable for continual graph representation learning since both the feature information and topological structure of the target node need to be considered appropriately.

More recently, Zhou et al. [39] proposed to store a set of representative experience nodes in a buffer and replay them along with new tasks (categories) to prevent forgetting existing tasks (categories). The buffer, however, only stores node features and ignores the topological information of graphs. Liu et al. [17] developed topology-aware weight preserving (TWP) that can preserve the topological information of existing graphs. However, its design hinders the capability of learning topology on new tasks (categories). Note that continual graph representation learning is essentially different from dynamic graph works which mainly concern time dependent graphs in which nodes and (or) edges change over time [37, 21, 40, 19]. Therefore, the methods developed for dynamic graphs cannot be directly applied to this task.

A desired learning system for continual graph representation learning is to continuously grasp knowledge from new categories of emerging nodes and capture their topological structures without interfering with the learned knowledge over existing graphs. To this end, we present a completely novel framework, *i.e.*, Hierarchical Prototype Networks (HPNs), to continuously extract different levels of abstract knowledge (in the form of prototypes) from graph data such that new knowledge will be accommodated while earlier experience can still be well retained. Within this framework, representation learning is simultaneously conducted to avoid catastrophic forgetting, instead of considering these two objectives separately. Specifically, based on the assumption that each node can be decomposed into basic atomic characteristics belonging to a set of attributes (*e.g.*, gender, nationality, hobby, *etc.*) and the relationship between a pair of nodes can be categorized into different types (*e.g.*, trust or distrust in a social network), we develop the Atomic Feature Extractors (AFEs) to decompose each node into two sets of atomic embeddings, *i.e.*, atomic node embeddings which encode the node feature information and atomic structure embeddings which encode its relations to neighboring nodes within multi-hop. Next, we present Hierarchical Prototype Networks to adaptively select, compose, and store representative embeddings with three levels of prototypes, *i.e.*, atomic-level, node-level, and class-level. Given a new node, only the relevant AFEs and prototypes in each level will be activated and refined, while others are uninterrupted. Eventually, each node can be represented with a tri-level prototypes which encode its feature as well as structure information from different abstract levels and can be used for downstream tasks such as node classification. Finally, we provide the theoretical analysis for the memory consumption upper bound of HPNs and its continual learning capability. To summarize, the main contributions of our work include:

- We present a novel framework, *i.e.*, Hierarchical Prototype Networks (HPNs), to continuously extract different levels of abstract knowledge (in the form of prototypes) from the graph data such that new knowledge will be accommodated while earlier experience can be well retained.

- We provide the theoretical analysis for the memory consumption upper bound of HPNs and its continual learning capability.

- Our experiment results on eight different public datasets demonstrate that the proposed HPNs not only achieve state-of-the-art performance, exhibiting good continual learning capability, but also use less parameters (more efficient). For instance, on OGB-Products dataset that contains more than 2 million nodes and 47 categories of nodes, HPNs achieves around 80% accuracy with only thousands of parameters.

## 2 Hierarchical Prototype Networks

In this section, we first state the problem we aim to study and the notations. Then we present Hierarchical Prototype Networks (HPNs) that consist of two core modules, *i.e.*, Atomic Feature Extractor (AFEs) and Hierarchical Prototype Networks (HPNs), as shown in Figure 1. AFEs serve to extract a set of atomic features from the given graph, and the HPNs aim to select, compose, and store the representative features in the form of different levels of prototypes. During the training stage, each node will only refine the relevant AFEs and prototypes of the model without interfering with the

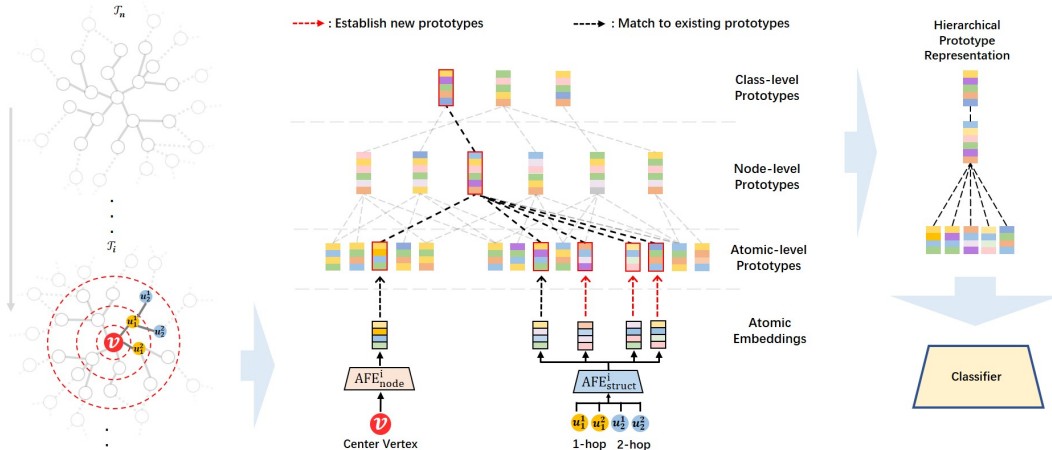

Figure 1: The framework of HPNs. On the left, subgraphs from different tasks come in sequentially. Given a node $v$. $u_k^j$ denotes the $j$-th sampled node from $k$-hop neighbors. In the middle, node $v$ and the sampled neighbors are fed into the selected AFEs to get atomic embeddings, which are either matched to existing A-prototypes or used as new A-prototypes. The selected A-prototypes are further matched to a N- and a C-prototype for the hierarchical representation, which is finally fed into the classifier to perform node classification.

irrelevant parts (*i.e.*, to avoid catastrophic forgetting). In the test stage, the model will activate the relevant AFEs and prototypes to perform the inference.

## 2.1 Problem Statement and Notations

We study continual learning on graphs that have new categories of nodes and associated edges (in the form of subgraphs) emerging in a continuous manner. In the context of continual learning, assuming we have a sequence of $p$ tasks $\{\mathcal{T}^i | i = 1, ..., p\}$, in which each task $\mathcal{T}^i$ aims to learn a satisfied representation for a new subgraph $\mathcal{G}_i$ consisting of nodes belonging to some new categories. A desired model should maintain its performance on all previous tasks after being successively trained on the sequence of $p$ tasks from $\mathcal{T}^1$ to $\mathcal{T}^p$.

For simplicity, we omit the subscripts in this section. Full notations will be used in the theoretical analysis. Each graph $\mathcal{G}$ consists of a node set $\mathbb{V} = \{v_i | i = 1, ..., N\}$ with $N$ nodes and an edge set $\mathbb{E} = \{(v_i, v_j)\}$ denoting the connections of nodes in $\mathbb{V}$. Each node $v_i$ can be represented as a feature vector $\mathbf{x}(v_i) \in \mathbb{R}^{d_v}$ that encodes node attributes, *e.g.*, gender, nationality, hobby, *etc*. The set of $l$-hop neighboring nodes of $v_i$ is defined as $\mathcal{N}^l(v_i)$, with $\mathcal{N}^0(v_i) = \{v_i\}$.

## 2.2 Atomic Feature Extractors

Based on the assumption that different nodes can be decomposed into basic atomic characteristics belonging to a set of attributes (*e.g.*, gender, nationality, hobby, *etc*.) and the relations between a pair of nodes can also be categorized into different types (*e.g.*, trust or distrust in a social network), we develop Atomic Feature Extractors (AFEs) to consider two different sets of atomic embeddings, *i.e.*, atomic node embeddings which encode the node features and atomic structure embeddings that encode its relations to neighbors within multi-hop. Specifically, to ensure that each node can be represented as different combinations of a subset of atomic features, AFEs are designed as learnable linear transformations $\text{AFE}_{\text{node}} = \{\mathbf{A}_i \in \mathbb{R}^{d_v \times d_a} | i \in \{1, ..., l_a\}\}$ and $\text{AFE}_{\text{struct}} = \{\mathbf{R}_j \in \mathbb{R}^{d_v \times d_r} | j \in \{1, ..., l_r\}\}$ where $\mathbf{A}_i$ and $\mathbf{R}_j$ are real matrices to encode atomic node and structure information, respectively. $l_a$ and $l_r$ denotes the cardinality of $\text{AFE}_{\text{node}}$ and $\text{AFE}_{\text{struct}}$, respectively. Given a node $v$, a set of atomic node embeddings is obtained by applying $\text{AFE}_{\text{node}}$ to the feature vector $\mathbf{x}(v)$:

$$\mathbb{E}_A^{\text{node}}(v) = \{\mathbf{x}^T(v)\mathbf{A}_i | \mathbf{A}_i \in \text{AFE}_{\text{node}}\}. \tag{1}$$

To obtain atomic structure embeddings, the multi-hop neighboring nodes of $v$ have to be considered. We first uniformly sample a fixed number of vertices from 1-hop up to $h$-hop neighborhood, *i.e.*, $\mathcal{N}_{sub}(v) \subseteq \bigcup_{l \in \{1,...,h\}} \mathcal{N}^l(v)$. Then these selected nodes are embedded via projection matrices in $\text{AFE}_{\text{struct}}$ to encode different types of interactions with the target node $v$:

$$\mathbb{E}_A^{\text{struct}}(v) = \{\mathbf{x}^T(u)\mathbf{R}_i | \mathbf{R}_i \in \text{AFE}_{\text{struct}}, u \in \mathcal{N}_{sub}\}. \tag{2}$$

Finally, the complete atomic feature set of target node $v$ is:

$$\mathbb{E}_A(v) = \mathbb{E}_A^{\mathrm{node}}(v) \cup \mathbb{E}_A^{\mathrm{struct}}(v). \tag{3}$$

Note that $\mathbf{A}_i$ and $\mathbf{R}_i$ are designed to generate different types of atomic features. To ensure that, we impose a divergence loss on AFEs to ensure they are be uncorrelated with each other and thus can map features to different subspaces:

$$\mathcal{L}_{div} = \sum_{i \neq j} \mathbf{A}_i^T \mathbf{A}_j + \sum_{i \neq j} \mathbf{R}_i^T \mathbf{R}_j. \tag{4}$$

## 2.3 Hierarchical Prototype Networks

With the atomic features extracted based on AFEs, hierarchical prototype networks (HPNs) will select, compose, and store representative features in the form of different levels of prototypes as shown in Figure 1. This is mainly achieved by refining existing prototypes and creating new prototypes only when necessary. Specifically, HPNs will produce three different levels of prototypes, *i.e.*, atomic-level prototypes (A-prototypes), node-level prototypes (N-prototypes), and class-level prototypes (C-prototypes). From atomic-level to class-level, the prototypes denote abstract knowledge of the graph at different scales which is analog to the feature maps of convolutional neural networks at different layers.

We first introduce how HPNs can refine existing prototypes. For each task that contains certain categories of nodes, instead of using all atomic embeddings generated by existing AFEs, HPNs only select a small and fixed number of AFEs from both $\mathrm{AFE}_{\mathrm{node}}$ and $\mathrm{AFE}_{\mathrm{struct}}$ which are more relevant to the given task. In this way, only the relevant AFEs are refined while others remain uninterrupted. Specifically, as shown in Figure 1, given a node from an incoming subgraph, each AFE is used to generate an embedding. Those AFEs with embeddings that are closer to existing A-prototypes are deemed as more confident ones and chosen. Formally, we first obtain $\mathbb{E}_A^{\mathrm{node}}(v)$ and $\mathbb{E}_A^{\mathrm{struct}}(v)$ via Eq. (1) and Eq. (2), respectively. Then, we calculate the maximum cosine similarity between atomic embeddings of each AFE ($\mathbf{e}_i$) and the A-prototypes as:

$$\mathrm{SimMAX}_i^{\mathrm{id}} = \max_{\mathbf{p}} (\frac{\mathbf{e}_i^T \mathbf{p}}{\|\mathbf{e}_i\|_2 \|\mathbf{p}\|_2}), \mathbf{e}_i \in \mathbb{E}_A^{\mathrm{id}}(v), \mathbf{p} \in \mathbb{P}_A, \tag{5}$$

where $\mathrm{id} \in \{\mathrm{node}, \mathrm{struct}\}$, $i$ ranges from 1 to $l_a$ (or $l_r$), and $\mathbb{P}_A$ is the atomic prototype set containing all A-prototypes. After that, we sort the AFEs in a descending order according to $\mathrm{SimMAX}_i^{\mathrm{id}}$ as $\mathrm{AFE}_{\mathrm{node}}^{\mathrm{sort}} = \{\mathbf{A}_{i'} \in \mathbb{R}^{d_v \times d_a} | i' \in \{1, ..., l_a\}\}$ and $\mathrm{AFE}_{\mathrm{struct}}^{\mathrm{sort}} = \{\mathbf{R}_{j'} \in \mathbb{R}^{d_v \times d_r} | j' \in \{1, ..., l_r\}\}$. Finally, we select the top $l_a'$ and top $l_r'$ ranked AFEs from these two sets as $\mathrm{AFE}_{\mathrm{node}}^{\mathrm{select}}$ and $\mathrm{AFE}_{\mathrm{struct}}^{\mathrm{select}}$, respectively. $l_a'$ and $l_r'$ are fixed hyperparameters with $l_a' \leq l_a$ and $l_r' \leq l_r$. The atomic embeddings generated by these selected AFEs are denoted as $\mathbb{E}_A^{\mathrm{select}}(v)$.

Based on $\mathbb{E}_A^{\mathrm{select}}(v)$, HPNs then starts to distill representative features, which is conducted by refining existing prototypes and creating new prototypes simultaneously. A matching process is first conducted between the $\mathbb{E}_A^{\mathrm{select}}(v)$ and $\mathbb{P}_A$ to recognize the atomic features that are compatible with exiting A-prototypes and those ones to be accommodated with new A-prototypes. Formally, we measure the cosine similarity between elements in $\mathbb{E}_A^{\mathrm{select}}(v)$ and elements in $\mathbb{P}_A$ as

$$\mathrm{Sim}_{E \to A}(v) = \{\frac{\mathbf{e}_i^T \mathbf{p}}{\|\mathbf{e}_i\|_2 \|\mathbf{p}\|_2} | \mathbf{e}_i \in \mathbb{E}_A^{\mathrm{select}}(v), \mathbf{p} \in \mathbb{P}_A\}. \tag{6}$$

The atomic embeddings that are compatible with existing A-prototypes are these ones with cosine similarity not less than a certain threshold $t_A$ to have at least one existing A-prototype, *i.e.*,

$$\mathbb{E}_{old}(v) = \{\mathbf{e}_i | \quad \exists \mathbf{p} \in \mathbb{P}_A \quad s.t. \quad \frac{\mathbf{e}_i^T \mathbf{p}}{\|\mathbf{e}_i\|_2 \|\mathbf{p}\|_2} \geqslant t_A\}. \tag{7}$$

$\mathbb{E}_{old}(v)$ collects a set of atomic embeddings satisfying the previous condition and can be used to refine $\mathbb{P}_A$. To this end, a distance loss $\mathcal{L}_{dis}$ is computed to enhance the cosine similarity between each $\mathbf{e}_i \in \mathbb{E}_{old}(v)$ and its corresponding A-prototype $\mathbf{p}_i \in \mathbb{P}_A$, *i.e.*,

$$\mathcal{L}_{dis} = -\sum_{\mathbf{e}_i \in \mathbb{E}_{old}(v)} \frac{\mathbf{e}_i^T \mathbf{p}_i}{\|\mathbf{e}_i\|_2 \|\mathbf{p}_i\|_2} \tag{8}$$

By minimizing $\mathcal{L}_{dis}$, not only the existing A-prototypes in $\mathbb{P}_A$ will get refined, the atomic embeddings will also be closer to 'standard' A-prototypes.

---
**Algorithm 1:** Learning Procedure for HPNs.
---
**Input** : Task sequence: $\{\mathcal{T}_1, ..., \mathcal{T}_p\}$, HPNs
1 **for** $\mathcal{T} \leftarrow 1$ **to** $p$ **do**
2    Get the data of the current task: $\mathbb{V}, \mathbb{E}, \mathbf{X}(\mathbb{V}) = \{\mathbf{x}(v)|v \in \mathbb{V}\}$.
3    Select $\text{AFE}_{node}^{select}$ and $\text{AFE}_{struct}^{select}$.
4    Compute $\mathcal{L} = \text{HPNs}(\mathbb{V}, \mathbf{X}(\mathbb{V}), \mathbb{E})$.
5    $\mathcal{L} = \text{HPNs}(\mathbb{V}, \mathbf{X}(\mathbb{V}), \mathbb{E})$.
6    Optimize $\mathcal{L}$.
---
**Output** : updated HPNs
---

Next, we discuss how to deal with the atomic embeddings that are not close to any existing prototypes, *i.e.*, $\mathbb{E}_{new}(v) = \mathbb{E}_A^{select}(v) \backslash \mathbb{E}_{old}(v)$ or $\mathbb{E}_{new}(v) = \{\mathbf{e}_i| \quad \forall \mathbf{p} \in \mathbb{P}_A, \quad \frac{\mathbf{e}_i^T \mathbf{p}}{\|\mathbf{e}_i\|_2 \|\mathbf{p}\|_2} < t_A\}$.

Contrary to $\mathbb{E}_{old}(v)$, atomic embeddings in $\mathbb{E}_{new}(v)$ are regarded as new atomic features of the corresponding AFEs. In this case, new prototypes should be generated to accommodate them. Considering that very similar embeddings may exist in $\mathbb{E}_{new}(v)$ and cause HPNs to create redundant prototypes, we first filter $\mathbb{E}_{new}(v)$ into $\mathbb{E}'_{new}(v)$ to keep only the representative ones such that

$$\forall \mathbf{e}_i, \mathbf{e}_j \in \mathbb{E}'_{new}(v), \frac{\mathbf{e}_i^T \mathbf{e}_j}{\|\mathbf{e}_i\|_2 \|\mathbf{e}_j\|_2} < t_A. \tag{9}$$

Then, $\mathbb{E}'_{new}(v)$ is included into $\mathbb{P}_A$ as new A-prototypes, which will be further refined in the future.

$$\mathbb{P}_A = \mathbb{P}_A \cup \mathbb{E}'_{new}(v). \tag{10}$$

After generating new prototypes, the matching will be conducted to get a new $\text{Sim}_{E \to A}(v)$ in which each element is not less than $t_A$. Then each element in $\mathbb{E}_A^{select}(v)$ is assigned a closest A-prototype according to $\text{Sim}_{E \to A}(v)$, and each node is associated with a set of atomic prototypes $\mathbb{A}(v)$.

To map $\mathbb{A}(v)$ to high level prototypes so as to obtain hierarchical prototype representations. $\mathbb{A}(v)$ is firstly mapped to a N-prototype denoting the overall features of $v$. We assume that N-prototypes lie in a $d_n$ dimensional space and a fully connected layer is applied to transform $\mathbb{A}(v)$ into the new space $\mathbb{E}_N(v) = \text{FC}_{A \to N}(\mathbf{a}_1 \oplus \cdots \oplus \mathbf{a}_{l'_a + l'_r}), \forall \mathbf{a}_i \in \mathbb{A}(v)$, where $\oplus$ denotes the concatenation operator. With $\mathbb{E}_N(v)$, we then find a matching N-prototype or establish a new one, which is similar to the process at atomic level except that the threshold is set as $t_N$, instead of $t_A$. Learning class-level prototypes from node-level prototypes is same except that we set the matching threshold as $t_C$. Finally, the hierarchical prototype representations of the target node is contained in the following set

$$\mathbb{P}_H(v) = \mathbb{A}(v) \cup \mathbb{N}(v) \cup \mathbb{C}(v). \tag{11}$$

Note that $\mathbb{A}(v)$ contains multiple A-prototypes denoting atomic features of $v$ from different aspects. $\mathbb{N}(v)$ and $\mathbb{C}(v)$ only contain one N-prototype and one C-prototype, representing the overall characteristics of $v$ and the common characteristics shared by the community containing $v$, respectively.

## 2.4 Learning Objective

The obtained hierarchical prototypes for each node are first concatenated into a unified vector and then pass through a fully connected layer FC to obtain a $c$ (the number of classes) dimensional feature vector, *i.e.*, $\text{FC}(\mathbf{h}_1 \oplus \cdots \oplus \mathbf{h}_{l'_a + l'_r + 2}), \forall \mathbf{h}_i \in \mathbb{P}_H(v)$. In this paper, we aim to perform node classification. Therefore, based on the $c$ dimensional feature vector and the softmax function $\sigma(\cdot)$, we can estimate the label with $\hat{y}_i = \sigma(\text{FC}(\mathbf{h}_1 \oplus \cdots \oplus \mathbf{h}_{l'_a + l'_r + 2}))_i$ where $i$ is the index of class. To perform node classification, with the output predictions $\hat{y}_i$ and the target label $y_i \in \{1, 2, ..., c\}$, the corresponding classification loss is given by

$$\mathcal{L}_{cls} = \sum_{i=1}^{c} -y_i \log(\hat{y}_i), \tag{12}$$

which is essentially the cross entropy loss function. Note that besides node classification, $\mathbb{P}_H(v)$ may also be used for other tasks based on different objective functions. In this paper, we focus on node classification and the overall loss of HPNs is:

$$\mathcal{L} = \mathcal{L}_{dis} + \mathcal{L}_{div} + \mathcal{L}_{cls}. \tag{13}$$

During the training stage, subgraphs with different tasks (containing different categories of nodes) are continuously fed to HPNs. Note that unlike topology-aware weight preserving (TWP) method [17], HPNs do not require task indicator for training and test, and therefore is more practical for real-world continual graph representation learning applications.

## 2.5 Theoretical Analysis

In this subsection, we provide the theoretical upper bound for the memory consumption and analyze how the model configuration would affect HPNs' capacity in dealing with different tasks. Both theoretical results are justified and analyzed in the experiments. Only the main results are provided here, while the detailed proof and analysis are given in Appendix.

We first show that the numbers of different prototypes are upper bounded by the number of atomic feature extractors and the dimension of the prototypes. Specifically, we have:

**Theorem 1** (Upper bounds for numbers of prototypes). *Given the notations defined in HPNs, the upper bound for the number of A-prototypes $n_a$ can be given by*

$$n_A \leqslant (l_a + l_r) \max_N S(d_a, N, 1 - t_A), \tag{14}$$

*and the upper bounds for the number of N-prototypes and the C-prototypes are:*

$$n_N \leqslant \max_N S(d_n, N, 1 - t_N) \quad \text{and} \quad n_C \leqslant \max_N S(d_c, N, 1 - t_C) \tag{15}$$

*where $S(n, N, t)$ is the spherical code defined on a $n$ dimensional hypersphere (details in Appendix).*

Theorem 1 provides an upper bound for the memory consumption of HPNs. In our experiments, we show that the number of parameters for most baseline methods are even higher than this upper bound.

Besides memory consumption, the more important problem for a continual learning model is the capability to maintain memory on previously learned tasks. Based on our model design, we formulate this as: whether learning new tasks affect the representations the model generates for old task data. We give explicit definitions on tasks and task distances based on set theory (in Appendix), then construct a bound to indicate what configuration would the model have to ensure this capability.

**Theorem 2** (Task distance preserving). *For* HPNs *trained on consecutive tasks $\mathcal{T}^p$ and $\mathcal{T}^{p+1}$. If $l_a d_a + l_r d_r \geqslant (l_r + 1) d_v$ and $\mathbf{W}$ is column full rank, then as long as $t_A < \lambda_{\min}(l_r + 1) \mathbf{dist}(\mathbb{V}_p, \mathbb{V}_{p+1})$, learning on $\mathcal{T}^{p+1}$ will not modify representations* HPNs *generate for data from $\mathcal{T}^p$, i.e. catastrophic forgetting is avoided.*

In Theorem 2, $\lambda_i$ is eigenvalues of the $\mathbf{W}^T \mathbf{W}$, where $\mathbf{W}$ is a matrix constructed via AFEs (details in Appendix). $d_v$, $d_a$ and $d_r$ are dimensions of data and two kinds of atomic embeddings. The bound in this theorem is not tight, as the tight bound would be dependant on the specific dataset properties. But this informs us that either the number of AFEs or the dimension of the prototypes has to be large enough to ensure that data from two tasks can be well separated in the representation space.

According to Theorem 1, the upper bound of the memory consumption is dependent on $S(d_a, N, t_A)$, $S(d_n, N, t_N)$, and $S(d_c, N, t_C)$. As $S(n, N, t)$ grows fast with $n$, we prefer larger number of AFEs with smaller prototype dimensions. We also empirically demonstrate this in Section 3.6. Besides, the upper bound proposed in Theorem 1 is explicitly computed and compared to experimental results.For both theorems, proofs and detailed explanations are included in Appendix.

## 3 Experiments

In the experiments, we answer the following six questions: (1) Whether HPNs can outperform state-of-the-art approaches? (2) How does each component of HPNs contribute to its performance? (3) Whether HPNs can memorize previous tasks after learning each new task? (4) Are HPNs sensitive to the hyperparameters? (5) Whether the theoretical results can be empirically verified? (6) Whether the learned prototypes can be interpreted via visualization?

### 3.1 Datasets

To assess the effectiveness of the proposed HPNs, we consider 8 datasets which include 3 citation networks (Cora [27], Citeseer[27], OGB-Arxiv [32, 20]), 3 web page networks (Wisconsin, Cornell, Texas) [22], 1 actor co-occurence network (Actor) [22], and 1 product co-purchasing networks (OGB-Products [4]). Detailed statistics about these datasets are provided in the Appendix.

Among these datasets, the results of 4 datasets, *i.e.*, Cora, Citeseer, OGB-Arxiv (169,343 nodes, 1,166,243 edges), and OGB-Products (2,449,029 nodes, 61,859,140 edges), are reported in the paper and the results of other 4 datasets are available in the Appendix.

### 3.2 Experimental Setup and Evaluation Metrics

To perform continual graph representation learning with new categories of nodes continuously emerging, we adopt a class-incremental scheme for all datasets. Each new task brings a subgraph with new categories of nodes and associated edges, *e.g.*, task 1 contains classes 1 and 2, task 2 contains

Table 1: Performance comparisons between HPNs and baselines on 4 different datasets.

| C.L.T. | Base | Cora | | Citeseer | | OGB-Arxiv | | OGB-Products | |
|---|---|---|---|---|---|---|---|---|---|
| | | AM/% | FM/% | AM/% | FM /% | AM/% | FM /% | AM/% | FM /% |
| None | GCN | 63.5±1.9 | -42.3±0.4 | 64.5±3.9 | -7.7±1.6 | 56.8±4.3 | -19.8±3.2 | 45.2±5.6 | -27.8±7.1 |
| | GAT | 71.9±3.8 | -33.1±2.3 | 66.8±0.9 | -19.6±0.3 | 54.3±3.5 | -21.76± 4.6 | 44.9±6.9 | -30.3±5.2 |
| | GIN | 68.3±2.3 | -35.4±3.4 | 57.7±2.3 | -36.4±0.3 | 53.2± 6.5 | -23.59 ±8.1 | 43.1±7.4 | -31.4±8.8 |
| EWC [13] | GCN | 63.1±1.2 | -42.7±1.6 | 54.4±4.2 | -30.3±0.9 | 72.1±2.4 | -9.1±1.9 | 66.7±0.5 | -8.4±0.4 |
| | GAT | 72.2±1.5 | -32.2±1.6 | 65.7±2.5 | -19.7±2.3 | 73.2 ±1.1 | -10.8 ±2.1 | 67.9±1.0 | -9.65±1.3 |
| | GIN | 69.6±2.6 | -28.5±2.8 | 57.9±3.4 | -36.3±2.4 | 74.1 ±1.7 | -8.3 ±2.0 | 67.3±2.3 | -13.6±1.5 |
| LwF [15] | GCN | 76.1±1.4 | -21.3±2.4 | 67.0±0.2 | -8.3±2.7 | 69.9 ± 3.9 | -12.1±2.8 | 66.3±2.5 | -11.8±3.4 |
| | GAT | 70.8±2.8 | -34.6±4.1 | 66.1±4.1 | -18.9±1.5 | 68.9±4.4 | -13.6±3.3 | 65.1±4.1 | -13.2±2.9 |
| | GIN | 74.1±2.7 | -23.3±0.8 | 63.1±1.9 | -16.5±2.2 | 71.4 ±4.8 | -15.9±5.6 | 65.9±4.0 | -10.7±3.1 |
| GEM [18] | GCN | 75.7±3.0 | -6.5±4.4 | 41.8±2.6 | -31.9±1.4 | 75.4±1.7 | -13.6±0.5 | 71.3±1.7 | -10.5±0.9 |
| | GAT | 69.8±3.0 | -26.1±2.6 | 71.3±2.2 | +9.0±1.5 | 76.6 ±0.7 | -11.3±0.4 | 70.4±0.8 | -10.9±1.6 |
| | GIN | 80.2±3.3 | -2.0±4.2 | 49.7±0.5 | -24.5±0.9 | 77.3 ±2.1 | -11.2±1.6 | 76.5±3.3 | -7.2±2.5 |
| MAS [1] | GCN | 65.5±1.9 | -21.4±3.7 | 59.5±3.1 | -0.1±2.4 | 69.8 ±0.4 | -18.8±0.9 | 62.0±1.1 | -17.9±1.9 |
| | GAT | 84.7±0.7 | -5.6±2.0 | 69.1±1.1 | -4.8±3.3 | 70.6 ±1.3 | -16.7 ±1.6 | 64.4±2.3 | -14.5±3.2 |
| | GIN | 76.7±2.6 | -4.0±3.6 | 65.2±3.9 | +0.0±2.0 | 65.3 ±2.9 | -17.0±2.3 | 61.4±3.8 | -20.9±2.9 |
| ERGN. [39] | GCN | 63.5±2.4 | -42.3±0.7 | 54.2±3.9 | -30.3±1.9 | 63.3±1.7 | -18.1±0.9 | 60.7±2.8 | -26.6±3.3 |
| | GAT | 71.1±2.5 | -34.3±1.0 | 65.5±0.3 | -20.4±3.9 | 63.5±2.4 | -19.5±1.9 | 61.3±1.7 | -25.1±0.8 |
| | GIN | 68.3±0.4 | -35.4±0.4 | 57.7±3.1 | -36.4±1.3 | 69.2± 1.8 | -11.8±1.4 | 61.8±4.7 | -23.4±7.9 |
| TWP [17] | GCN | 68.9±0.9 | -5.7±1.5 | 60.5±3.8 | -0.3±4.4 | 75.6±0.3 | -10.4±0.5 | 69.9±0.4 | -9.0±1.1 |
| | GAT | 81.3±3.2 | -14.4±1.5 | 69.8±1.5 | -8.9±2.6 | 75.8±0.5 | -5.9±0.3 | 69.3±2.3 | -8.9±1.5 |
| | GIN | 73.7±3.2 | -3.9 ±2.6 | 68.9±0.7 | -2.4±1.9 | 76.6±1.8 | -11.3±1.1 | 69.9±1.4 | -10.3±2.7 |
| Join. | GCN | 93.7 ± 0.5 | 0.0±0.0 | 78.9 ± 0.4 | 0.0±0.0 | 77.2±0.8 | 0.0±0.0 | 72.9±1.2 | 0.0±0.0 |
| | GAT | 93.9 ± 0.9 | 0.0±0.0 | 79.3 ± 0.8 | 0.0±0.0 | 81.8±0.3 | 0.0±0.0 | 73.7±2.4 | 0.0±0.0 |
| | GIN | 93.2 ± 1.2 | 0.0±0.0 | 78.7 ± 0.9 | 0.0±0.0 | 82.3±1.9 | 0.0±0.0 | 77.9±2.1 | 0.0±0.0 |
| **HPNs** | | **93.7±1.5** | **+0.6±1.0** | **79.0±0.9** | **-0.6±0.7** | **85.8± 0.7** | **+0.6±0.9** | **80.1±0.8** | **+2.9±1.0** |

Figure 2: (a) and (b) are AM and FM of HPNs with different number of AFEs and prototype dimensions on OGB-Arxiv. (c) and (d) are AM and FM change with when $t_A$ varies on Cora.

new classes 3 and 4, *etc*. Each model is trained on a sequence of tasks, and the performance will be evaluated on all previous tasks. Specifically, we adopt accuracy mean (AM) and forgetting mean (FM) as metrics for evaluation. After learning on all tasks, the AM and FM are computed as the average accuracy and the average accuracy decrease on all previous tasks. Negative FM indicates the existence of forgetting , zero FM denotes no forgetting and positive FM denotes positive knowledge transfer between tasks. For HPNs, we set $d_a = d_n = d_c = 16$, $l_a = l_r = 22$, and $h = 2$. The threshold $t_A$, $t_N$, and $t_C$ are selected by cross validation on $\{0.01, 0.05, 0.1, 0.15, 0.2, 0.25, 0.3, 0.35, 0.4\}$. The experiments on the important hyperparameters are provided in Section 3.6. All experiments are run on an Nvidia Titan Xp GPU. Full implementation details are in Appendix, and the code is available in supplementary materials.

### 3.3 Comparisons with Baseline Methods

We compare HPNs with various baseline methods. Experience Replay based GNN (ERGNN) [39] and Topology-aware Weight Preserving (TWP) [17] are developed for continual graph representation learning. The others approaches, including Elastic Weight Consolidation (EWC) [13], Learning without Forgetting (LwF) [15], Gradient Episodic Memory (GEM) [18], and Memory Aware Synapses (MAS) [1] are popular continual learning methods for Euclidean data. All the baselines are imple-

Table 2: Ablation study on prototypes of different levels of prototypes over Cora.

| Conf. | A-p. | N-p. | C-p. | AM% | FM% |
|---|---|---|---|---|---|
| 1 | ✓ | | | 89.2±1.3 | -0.1±0.5 |
| 2 | ✓ | ✓ | | 91.7±1.1 | -0.2±0.8 |
| 3 | ✓ | ✓ | ✓ | 93.7±1.5 | +0.6±1.0 |

Table 3: Ablation study on different loss terms over Cora.

| Conf. | $\mathcal{L}_{cls}$ | $\mathcal{L}_{div}$ | $\mathcal{L}_{dis}$ | AM% | FM% |
|---|---|---|---|---|---|
| 1 | ✓ | | | 92.4±1.3 | +0.8±0.7 |
| 2 | ✓ | ✓ | | 92.9±1.1 | +0.3±1.0 |
| 3 | ✓ | | ✓ | 92.8±0.9 | +0.0±1.2 |
| 4 | ✓ | ✓ | ✓ | 93.7±1.5 | +0.6±1.0 |

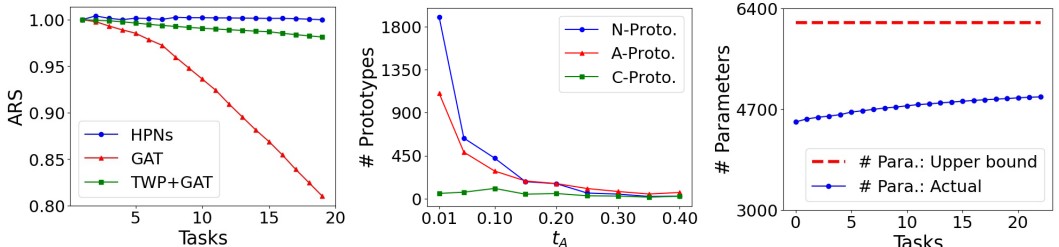

Figure 3: Left: dynamics of ARS for continual learning tasks on OGB-Arxiv. Middle: impact of $t_A$ on the number of prototypes in HPNs over Cora. Right: dynamics of memory consumption of HPNs on OGB-Products.

Table 4: Final parameter amount for models trained on OGB-Products

|  | None | EWC | LwF | GEM | MAS | ERGNN | TWP | Joint | HPNs |
|---|---|---|---|---|---|---|---|---|---|
| GCN | 2,336 | 46,720 | 4,672 | 2,202,336 | 2,336 | 6,738 | 9,344 | 2,336 | |
| GAT | 20,032 | 400,640 | 40,064 | 2,220,032 | 20,032 | 24,432 | 80,128 | 20,032 | 4,908 |
| GIN | 2,352 | 47,040 | 4,704 | 2,202,352 | 2,352 | 6,752 | 9,408 | 2,352 | |

mented based on three popular backbone models, *i.e.*, Graph Convolutional Networks (GCNs) [12], Graph Attentional Networks (GATs) [31], and Graph Isomorphism Network (GIN) [34].

Note that Joint training (Join.) in Table 1 does not represent continual learning. It allows a model to access data of all tasks at any time and thus is often used as an upper bound for continual learning.[29].

In Table 1, we observe that regularization based approaches, *e.g.*, EWC and TWP, generally obtain lower forgetting, but the accuracy (AM) is limited by the constraints. However, the forgetting problem of regularization based methods will become increasingly severe when the number of tasks is relatively large, as shown in Section 3.5. Memory replay based methods such as GEM achieve better performance without using any constraint. However, the memory consumption is higher (Section 3.7). HPNs significantly outperform all baselines without inheriting their limitations. Compared to regularization based methods, HPNs do not impose constraints to limit the model's expressiveness, therefore the performance is much better. Compared to memory replay based methods, HPNs do not only perform better but also are memory efficient as shown in Section 3.7. Joint training (Join.) achieves comparable performance to HPNs on small datasets but is significantly worse on large OGB datasets. This is because joint training (Join.) is a multi-task setting, inter-task interference may cause negative transfer, which is not obvious on small datasets with only a few tasks but becomes prominent on large datasets with tens of tasks. In HPNs, different tasks can choose different combinations of the parameters and thus task interference is dramatically alleviated.

### 3.4 Ablation Study

We conduct ablation studies on different levels of prototypes and different combinations of three loss terms. In Table 2, we show the performance of HPNs when A-, N-, and C-Prototypes are gradually added (Cora dataset). We notice both AM and FM of HPNs increase when higher level prototypes are considered. This suggests that high level prototypes can enhance the model's performance and robustness against forgetting.The effect of different combinations of loss terms are shown in Table 3. The first three rows show that adding $\mathcal{L}_{div}$ or $\mathcal{L}_{dis}$ with $\mathcal{L}_{cls}$ may slightly improve the performance. By jointly considering these three terms, the performance (AM) can be further improved. This is because $\mathcal{L}_{div}$ pushes different AFEs away from each other and $\mathcal{L}_{dis}$ makes the prototypes of each AFE be more close to its output. Jointly considering $\mathcal{L}_{div}$ and $\mathcal{L}_{dis}$ with $\mathcal{L}_{cls}$ can make the prototype space better separated as shown in Section 3.8.

### 3.5 Learning Dynamics

For continual learning, it is important to memorize previous tasks after learning each new task. To measure this, instead of directly measuring the average accuracy on previous tasks which may mix up the accuracy change caused by forgetting and task differences, we develop a new metric, *i.e.*, average retaining score (ARS), to address this problem. Specifically, after learning on a task $\mathcal{T}^i$, the ratio between the model's accuracy on a previous task $\mathcal{T}^{i-m}$ and its accuracy on $\mathcal{T}^{i-m}$ after it had been just learned on $\mathcal{T}^{i-m}$ is defined as the retaining ratio. Then the ARS is the average retaining ratio of all previous tasks after learning a new task.

Figure 3(left) shows the ARS change of HPNs and two baselines. GAT represents the models without continual learning techniques. TWP+GAT is the best baseline in terms of forgetting. GAT forgets quickly, while TWP significantly alleviates the forgetting problem for GAT. But as more tasks come in, the forgetting of TWP+GAT increases. As different tasks require different parameters, TWP+GAT

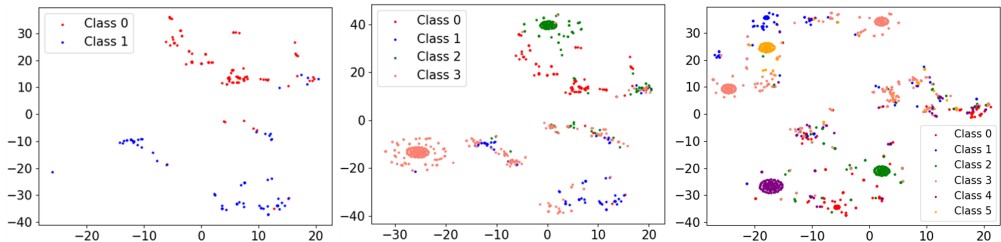

Figure 4: Visualization of hierarchical prototype representations of nodes in the test set of Cora.

(regularization based) is seeking a trade off between old and new tasks. With more new tasks, TWP+GAT tends to gradually adapt to new tasks and forget old ones. On contrary, HPNs maintain the ARS very well. This is because HPNs learn prototypes to denote the common basic features and learning new tasks does not hurt the parameters for old tasks. New tasks can be handled with new combinations of the existing basic prototypes. If necessary, new prototypes can be established for more expressiveness.

### 3.6 Parameter Sensitivity

As discussed in Section 2.5, the number of AFEs and the prototype dimensions are key factors in determining the continual learning capability and memory consumption. Here, we conduct experiments with different number of AFEs and prototype dimensions to justify the theoretical results. We keep the dimensions of different prototypes equal and the number of two types of AFEs equal for simplicity.

As shown in Figure 2(a) and (b), larger dimensions and the number of AFEs yield better AM and FM, which is consistent with Theorem 2. Besides, AM is mostly determined by the number of AFEs since HPNs compose prototypes with different AFEs to represent each target node. The number of possible combinations determines its expressiveness. Considering the above results and the bound (Theorem 1) for the number of prototypes, using large number of AFEs and small dimension can ensure both high performance and low memory usage, as verified in Section 3.7.

We also evaluate the effectiveness of HPNs when prototype thresholds vary from 0.01 to 0.4. Here, we set $t_A = t_N = t_C$ for simplicity. In Figure 2(c) and (d), we observe that the performance (AM and FM) of HPNs are generally stable when $t_A$ varies and slightly better when $t_A$ is between 0.2 and 0.3. This is because when $t_A$ is too small or too large, we will have too many or too less prototypes (consistent with Theorem 1) as shown in Figure 3(middle), which may cause the problem of overfitting or underfiting.

### 3.7 Memory Consumption

We compare memory consumption of different methods, as well as a explicitly theoretical memory upper bound, with the baselines on OGB-Products (the largest dataset). We also show the actual memory consumption of HPNs in the process of continual learning.

In Table 4, even on the dataset with millions of nodes and 23 tasks, HPNs can accommodate all tasks with a small amount of parameters. Besides, the dynamic change of parameter amount is shown in Figure 3(right). The red dashed line denotes the theoretical upper bound (6,163), and the computation details are included in Appendix. In Figure 3(right), we notice the actual memory usage of HPNs is much lower than the upper bound. Moreover, even the upper bound is among the lowest for memory consumption compared to baselines. The model we use here is the same as the one in Section 3.3

### 3.8 Visualization

To show that HPNs can generate interpretable prototype representations, we apply t-SNE [30] to visualize the node representations of the Cora dataset (test set) after learning each task. As shown in Figure 4, each task contains two classes corresponding to (red, blue), (green, salmon), and (purple, orange), as new tasks come in gradually, the representations are consistently well separated, which will be beneficial for downstream tasks.

## 4 Conclusion

In this paper, we proposed Hierarchical Prototype Networks (HPNs), to continuously extract different levels of abstract knowledge (in the form of prototypes) from streams of tasks on graph representation learning. The performance of HPNs is both theoretically and experimentally justified. In the future, we will apply HPNs to more application scenarios like link prediction, multi-label classification, anomaly detection, *etc*.

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
