# OpenReview forum: "Hierarchical Prototype Network for Continual Graph Representation Learning"
_NeurIPS.cc/2021/Conference — NeurIPS 2021 Submitted_

### Official Review · Reviewer_eRkd · 2021-07-12

**Rating:** 6
**Confidence:** 3

**Summary:**

This paper introduces continual learning in graph settings, where new graph-based tasks come in over time. To learn effective node classifiers for new tasks without forgetting the old tasks, a three-level prototype network is outlined. Atomic, node, and class prototypes are jointly used on top of Atomic Feature Extractors, where only the relevant features and prototypes are selected and updated in each new task, without disturbing the others. Experiments on eight benchmarks show the effectiveness and lack of forgetting for continual node classification.

**Limitations And Societal Impact:**

The main limitation that is discussed is that only node classification is investigated. The checklist states that there are no negative societal impacts of the work.

**Main Review:**

[Strengths]

This paper addresses a challenging task in graph learning, namely the scenario where new graph tasks come in over time. The proposed multi-level approach is interesting at a high level as it addresses different levels of granularity.

The paper furthermore excels at outlining current advances in continual learning and specifically in the graph context. This outline helps not only with providing the proper background and starting point of this paper, but also helps to understand what is new here.

The presented results show that the approach not only obtains a high accuracy overall, but at the cost of minimal loss of accuracy with the introduction of new tasks. These results highlight the potential of the proposed approach.

[Weaknesses]

There are a number of limitations to the paper that require addressing.

First, the method is not written well and unnecessarily hard to go through. Section 2 contains many unintuitive steps and unclear notations that make it hard to understand what is going on. For example:
- What is the idea behind subsampling the neighbourhood in lines 125-127?
- According to Equation 3, the node v is represented by a set of features, both from node and structure information. Why are these two sets combined in Equation 3 and separated again in Equation 5? And how do you go from AFE_{node}^{select} and AFE_{struct}^{select} to \mathbb{E}_A^{select}(v)?  There is no explanation and completely unclear.
- What is the point of Algorithm 1? It contains so few steps that there is no information anymore. Steps 4 and 5 are the same too.
- Where does \mathbb{A}(v) magically come from in line 178? What does associated with mean? And how big is the set  for each node? Again not clear from the method section.
- The N-prototypes are obtained from \mathbb{A}(v), and C-prototypes are obtained from N-prototypes. What does Equation 11 mean in that case? How can you throw them on one pile?
- At the end, there is a cross-entropy objective in Equation 12, where the likelihoods are obtained from a fully-connected layers that takes inputs from all levels of prototypes. Don’t the prototypes have different dimensionalities however? What is going on on lines 193 and 195?

Next to the mathematical explanation, the writing also requires improvement. A few examples from Section 2.4 and 2.5 (not a full list):
- Line 192: pass = passed
- Line 203: unlike topology-aware = unlike the topology-aware
- Line 204: HPNs do not require task indicator = HPNs not require a task indicator
- Line 228: is = denotes the / are the

Section 2.5 provides two theorems, but the appendix only provides a proof for Theorem 2. Where is the proof for Theorem 1?

In the experiments, Table 1 shows that the proposed approach outperforms all other approaches. Where do the numbers of the baselines come from? Baseline [39] should btw be updated to:

[a] Zhou, Fan, and Chengtai Cao. "Overcoming Catastrophic Forgetting in Graph Neural Networks with Experience Replay." Proceedings of the AAAI Conference on Artificial Intelligence. Vol. 35. No. 5. 2021.

In [a], they report a best PM of 95.66% and 81.83% on Cora and Citeseer, compared to 93.7% and 79.0% for this paper. Notably, [39] obtains only 71.1% and 65.5%. How come there are such big differences between what is reported in the original paper and what is reported in Table 1? Table 1 suggest that the proposed approach is far better than the baselines, while a direct comparison between papers suggests otherwise.

[Overall]

This paper addresses a challenging task with a multi-level approach. The method is intuitive at a high level and the related work is outlined well. The method section itself can however hardly be followed and requires big changes. Experimentally, there are also open questions that require attention regarding reported numbers.

**Time Spent Reviewing:**

4

---

> ### Author Response · Authors · 2021-08-10
> **Response to Reviewer eRkd**
>
> We sincerely thank the reviewer for the recognition of our contributions and the careful checking on the details. We will carefully improve the clarity of the confusing parts mentioned in the reviews. Detailed responses are provided below.
>
> Q1. What is the idea behind subsampling the neighbourhood in lines 125-127?
>
> A1:  Similar to  GraphSAGE [d], we use the uniform sampling strategy on the neighbourhood. The aim is to make the model scalable on large graphs. As mentioned in [d], the number of neighbours is unpredictable for each graph node, without subsampling, the expected runtime of each batch would be unpredictable and in the worst case $\mathcal{O}(|\mathcal{V}|)$, where $|\mathcal{V}|$ is the number of nodes in the graph. Besides, for our model, subsampling the neighbourhood ensures that all nodes get the same number of structure embeddings, which makes it possible to stack embeddings of multiple nodes into one tensor for parallel computation.
>
> Q2. Why are the two sets combined in Equation 3 and separated in Equation 5?
>
> A2. Equation 3 serves to denote that the embeddings generated for a node $v$ (i.e. $\mathbb{E}\_A(v)$) contains two subsets $\mathbb{E}\_A\^{\textrm{node}} (v)$ and $\mathbb{E}\_A\^{\textrm{struct}}(v)$. In Equation 5, we treat these two sets separately since in the following sorting procedure with the AFEs, the AFEs for node and structure are sorted separately and need separate indices (node or struct) for clarity.
>
> Q3. How to go from $\mathrm{AFE}\_{\textrm{node}}^{\textrm{select}}$ and $\mathrm{AFE}\_{\textrm{struct}}^{\textrm{select}}$ to $\mathbb{E}^{\textrm{select}}\_A(v)$?
>
> A3. This is same as the computation from Equation 1 to 3 except the $\mathrm{AFE}\_{\textrm{node}}$ and $\mathrm{AFE}\_{\textrm{struct}}$ are replaced by $\mathrm{AFE}\_{\textrm{node}}^{\textrm{select}}$ and $\mathrm{AFE}\_{\textrm{struct}}^{\textrm{select}}$ so we omitted that due to space limitations. We will add the detailed computation in the Appendix for clarity. Specifically, the $\mathrm{AFE}\_{\textrm{node}}^{\textrm{select}}$ and $\mathrm{AFE}\_{\textrm{struct}}^{\textrm{select}}$ are used to generate two embedding sets $\mathbb{E}^{\textrm{node,select}}\_A(v)$ ={$\mathbf{x}^T(v)\mathbf{A}\_i  | \mathbf{A}\_i\in\mathrm{AFE}\_{\textrm{node}}^{\textrm{select}}$} and $\mathbb{E}^{\textrm{struct,select}}\_A(v)$ =  {$\mathbf{x}^T(u)\mathbf{R}\_i | \mathbf{R}\_i\in\mathrm{AFE}\_{\textrm{struct}}^{\textrm{select}}, u\in \mathcal{N}_{sub}$}, like Equation 1 and 2. Then the union of these two set is denoted as $\mathbb{E}\_A^{\textrm{select}}(v) = \mathbb{E}^{\textrm{node,select}}\_A(v) \cup \mathbb{E}^{\textrm{struct,select}}\_A(v)$, like Equation 3.
>
> Q4. What is the point of Algorithm 1?
>
> A4. We formulate Algorithm 1 to give a concise overview of the whole training procedure, as the method part contains many details and may cause confusion about the training process.
> Step 4 and 5 are indeed repeated, and we will correct that.
>
> Q5. Where does $\mathbb{A}(v)$ come from in line 178, what does 'associated' mean, and how big is the set?
>
> A5. $\mathbb{A}(v)$ is obtained by selecting the closest A-prototype for each element in the embedding set $\mathbb{E}_A^{\textrm{select}}(v)$. And the closeness is measured by the just computed similarity $\mathrm{Sim}\_{{E\rightarrow A}}(v)$ (Eq. 6, Section 2.3 in paper, line 161, page 4). Therefore, the size of $\mathbb{A}(v)$ is same as size of $\mathbb{E}_A^{\textrm{select}}(v)$, i.e. $l\_a'+l\_r'$, the number of atomic embeddings generated for node $v$. Finally, the 'associated with' in line 178 means that the selected A-prototypes (i.e. $\mathbb{A}(v)$) are used for the representations of node $v$.
>
> Q6. What does Equation 11 mean?
>
> A6. Equation 11 is not a computation operation. Instead, It denotes that the prototypes selected for the node $v$ (i.e. $\mathbb{P}\_H(v)$) includes the prototypes from three levels ($\mathbb{A}(v)$, $\mathbb{N}(v)$, and $\mathbb{C}(v)$) (i.e., union of the three subsets).
>
> Q7. In Equation 12, don’t the prototypes from all levels have different dimensionalities however?
>
> A7. Yes, different level prototypes may have different dimensions. But they are concatenated into one vector. More concretely, suppose we have only three vectors $v_a \in \mathbb{R}^{d\_a}$, $v\_n \in \mathbb{R}^{d\_n}$, and $v\_c \in \mathbb{R}^{d\_c}$ denoting three prototypes from the three different levels, respectively. Before feeding them into the fc layer, they are concatenated into one vector $v \in \mathbb{R}^{d\_a+d\_n+d\_c}$ (line 193, page 5, Section 2.3 in paper). And the input dimension of the fc layer is $d\_a+d\_n+d\_c$.
>
> Q8. About improving the writing.
>
> A8. We will carefully proofread the paper and correct these typos and grammar errors.
>
> Q9. Where is the proof for Theorem 1?
>
> A9. The proof of Theorem 1 is provided in the Section 1.2 of the Appendix (line 14~42, page 1). The analysis following the definition of spherical code explains how the upper bound is given. The bound can only be represented as an expression containing the spherical code $S(n, N, t)$, as there is currently no general formulation of $N$ for an arbitrary $n$. Nevertheless, we give the explicit formulation for the 2-dimension case, which is also validated in the experiments (Section 3.7 in paper, line 343, page 9).
>
> Q10. Where do the numbers of the baselines come from?
>
> A10. The results are obtained by running the baselines on the same experimental settings. All the baseline codes (except the ERGNN [a]) are implemented and publicly released by the authors of [b]. We also fixed some baselines that are not directly runnable. As for the ERGNN model, we also implemented it based on the continual learning framework provided by the authors of [b]. For all of the baselines, we used the same cross validation to tune the hyperparameters so as to obtain the best model configuration as we explained in the appendix. (Section 2.2 in Appendix, line 274\~278, page 7)
>
> Q11. How come there are such big differences between what is reported in the original paper [a] and what is reported in Table 1?
>
> A11. The reason of the inconsistent performance reported in our paper and in [a] is twofold:
>
> First, different metrics were used. As explained in [a], they defined performance mean (PM) as the average of the accuracy a model obtains after learning each task. This metric does not reflect the accuracy decrease (forgetting) caused by training the model on the subsequent tasks. Differently, our accuracy mean (AM) is the average of accuracy on each task after the model has been sequentially trained on all tasks (Section 3.2, line257~259, page 7). This metric is widely adopted in nearly all continual learning works [b, c] as it reflects the accuracy decrease on previous tasks after the model is trained on subsequent tasks (i.e. how much the model forgets about previous works).
>
> More concretely, suppose a model is trained on a sequence of tasks, after learning each task $\mathcal{T}\_i$, the model has an accuracy $acc_i$ on $\mathcal{T}\_i$. After being trained on all the tasks, the model is again tested on all previous tasks and obtains new accuracies $acc_i'$ on each task $\mathcal{T}\_i$. The PM defined in [a] is the average of $acc_i$, while the AM in our work is the average of $acc_i'$. Therefore, the PM in [a] would no doubt give higher values as it does not count in the forgetting problem.
>
> The results reported in the left part of Table 3 in Appendix (line 320, page 9) can serve as a real example. It shows the performance of the pure GAT model (code is publicly available) on the Cora dataset. Calculating the PM from [a] is to get the average of the diagonal entries of the matrix. The result is (94.12%+93.3%+94.4%)/3 = 93.94%, which is very close to the 94.19% reported in [a]. And calculating the AM in our paper would be the average over the entries in the third column, and the result would be (71.49%+49.68%+94.44%)/3 = 71.87%.
> Similar results can also be obtained for other baseline models.
>
> Second, the task split and the train-test split may also be a reason for the inconsistency. The task split and train-test split is not provided in [a] and their code is not publicly available yet. Our task split and the train\-test split was reported in Appendix (Section 2.1 in Appendix, line 182\~243, page 5). We adopted the default train-test split provided by the authors of the datasets, and the task split was designed to minimize the class imbalance within each task (line 211\~213, line 240\~242, page 6, in Appendix). Besides, the metric definition, task splitting, and train-test splitting of the concerned Cora and Citeseer datasets can be found in our submitted code. We will also release our code to the public for reproducibility.
>
> Q12. Baseline [39] should be updated.
>
> A12. Thanks for pointing out, we will correct this.
>
> Q13. Only node classification is investigated.
>
> A13. In this work, we aim to address the problem of continual graph representation learning in which new categories of nodes and their associated edges are continuously emerging. Based on the promising node classification performance, we will extend our work to other tasks such as link prediction and graph classification in future works.
>
> [a] Zhou, Fan, and Chengtai Cao. "Overcoming Catastrophic Forgetting in Graph Neural Networks with Experience Replay." Proceedings of the AAAI Conference on Artificial Intelligence. Vol. 35. No. 5. 2021.
>
> [b] Liu, H., Yang, Y. and Wang, X., 2020. Overcoming catastrophic forgetting in graph neural networks. arXiv preprint arXiv:2012.06002.
>
> [c] Lopez-Paz, D. and Ranzato, M.A., 2017. Gradient episodic memory for continual learning. Advances in neural information processing systems, 30, pp.6467-6476.
>
> [d] Hamilton, W.L., Ying, R. and Leskovec, J., 2017, December. Inductive representation learning on large graphs. In Proceedings of the 31st International Conference on Neural Information Processing Systems (pp. 1025-1035).

---

> > ### Comment · Reviewer_eRkd · 2021-09-01
> > **End of rebuttal**
> >
> > The additional information has helped me to understand where the numbers come from and has helped to understand parts of the method. Due to the divering scores, we have discussed the paper amongst reviewers. After the discussion, I remain at a 6. The high-level idea remains interesting to me. After the rebuttal, there are still some concerns about the clarity of the method which require attention.

---

> > > ### Author Response · Authors · 2021-09-01
> > > **Thanks for the support**
> > >
> > > Thanks a lot for your support and the constructive comments. We also promise to improve the clarity in the updated version.

---

### Official Review · Reviewer_eQnQ · 2021-07-15

**Rating:** 7
**Confidence:** 4

**Summary:**

This paper aims to address the catastrophic forgetting issue with continual learning on streaming graphs. To comprehensively incorporate the feature and topological information of new nodes, this paper proposes atomic feature extractors to project raw node features and neighbor nodes into distinct base embeddings, which are then used to refine or compose atomic-, node-, and class-level prototype embeddings in a hierarchical manner. Experiments demonstrate that by doing so, catastrophic forgetting can be avoided and promising accuracy can be achieved on continual graph learning tasks.

**Limitations And Societal Impact:**

The limitations of this paper are mainly associated with the lack of detailed explanations w.r.t. some technical claims, and the authors are recommended to provide more corresponding justifications.

**Main Review:**

The highlights of this paper are as follows:
1. The proposed approach is well motivated, where the compositional node embeddings based on hierarchical prototypes is able to make the approach scalable to larger datasets.
2. Despite some syntax errors, the presentation of this paper is overall clear and easy to follow.
3. The experimental evaluations are thorough, spanning across a wide range of heterogeneous graph datasets, and most technical claims are backed by the results.

At the same time, several shortcomings of this paper are also noticed, which are listed below:
1. The parameter-saving claim only holds when the input node features do not include node IDs, where the AFEs only need to perform transformations on the low-dimensional feature vectors. In some other graph learning settings where node IDs are used as the input feature, AFEs will essentially be multiple node embedding tables where $d_v$ is the total number of nodes, which will dominate the parameter size and become less scalable than other methods that only maintain one node embedding table.
2. Equations 5,6,7,8 need 4 repetitive computations of pairwise cosine similarity between atomic features and the prototype embeddings, which can be very time-consuming given the potential size of set $\mathbb{P}$. Is the approach able to reuse results from previous steps (e.g., Eq 5) to cut the time cost?
3. The atomic features are refined by a hard selection process with pre-defined threshold. While the learning process seems to be end-to-end in Algorithm 1, it remains unclear how the hard selection of atomic features is made differentiable to allow for end-to-end training.
4. The overall learning objective directly combines three loss functions without any coefficients to balance any part of the sub-losses, which is uncommon. It is expected to provide more details and justifications on doing so.
5. As mentioned above, several syntax errors are found in this paper:
- Incomplete sentences: Figure 1 --> “Given a node v”; Section 2.3 --> “To map … prototype presentations.”;
- Section 2.2., “they are be uncorrelated” --> are uncorrelated
- Section 2.3, “HPNs then starts to” --> start
- Section 2.3, “a N-prototype” --> an

In summary, the highlights of this paper outweighs its shortcomings, while this paper can still benefit from some clarifications on the issues listed above.


**Time Spent Reviewing:**

4

---

> ### Author Response · Authors · 2021-08-10
> **Response to Reviewer eQnQ**
>
> We sincerely thank the reviewer for the support and constructive comments. We will carefully take the reviews, especially the ones on explanations of the technical claims, to further improve our manuscript. Following are the detailed responses to each question.
>
>
> Q1. The parameter-saving claim only holds when the input node features do not include node IDs.
>
> A1: Thanks for raising this concern. Actually, maintaining multiple embedding tables does not cause a problem on the scalability, as the embedding dimension of our model could be relatively small. According to experiment description in Section 2.2 of the Appendix (line 270, page 7), even on the large OGB datasets with rich node features, using relatively small embedding dimensions can still guarantee very good performance. Overall, although our model maintains multiple embedding tables when nodeID is used, using small embedding dimensions can ensure the total memory usage be feasible.
>
> Q2. Is the approach able to reuse results from previous steps (e.g., Eq 5) to cut the time cost?
>
> A2:  Yes, the computation of the similarity in Eq. 5,6,7,8 is not repeated in practice, and is parallelized by being implemented with matrix multiplication.
> Besides, the size of $\mathbb{P}$ is bounded by our Theorem 1 (Section 2.5 in paper, line 312\~223, page 6).
> Experimentally, even on large datasets with millions of nodes, the size of $\mathbb{P}$ is still well controlled (Section 3.7 in paper, line 341\~346, page 9; Section 3.5 in Appendix, line 325\~331, page 10),
>
> Q3. It remains unclear how the hard selection of atomic features is made differentiable to allow for end-to-end training.
>
> A3: The hard selection process is non-differentiable and is used as a preprocessing step for each batch. We make the process differentiable by using the distance loss in Equation 8 (Section 2.3 in paper, line 166, page 4). After the selection process, the matched prototypes and the embeddings are pushed close to each other by Equation 8, which will both refine the prototypes and the embeddings.
>
>
> Q4. The overall learning objective directly combines three loss functions without any coefficients to balance any part of the sub-losses, which is uncommon.
>
> A4: Our earlier results suggest that the model (HPNs) performance is not very sensitive to the coefficients for $\mathcal{L}\_{div}$ and $\mathcal{L}\_{dis}$ when these two coefficients varies. For simplicity, we choose to omit the coefficients in the paper. Specifically, below is the results of adding two coefficients on the two auxiliary losses, i.e. $\mathcal{L} = \mathcal{L}\_{cls} + \alpha \cdot \mathcal{L}\_{div} + \beta  \cdot \mathcal{L}\_{dis} $. We iteratively fix each of these hyperparameters and tune the other one to check the parameter sensitivity. The first two tables are on the Cora dataset, and the third and fourth are on the Citeseer dataset. When tuning $\alpha$, we fix $\beta=1$, and vice versa.
>
> * $\beta=1$, on Cora dataset
>
> |$\alpha$|0.0001|0.001|0.01| 0.1 |1.0 |10|100|1000|10000|
> |---|---| ---|---|---| ---| ---|---|---|---|
> |AM/%|92.3|92.6|92.3| 93.1 |93.8 | 93.5| 93.1| 93.2|81.5|
> |FM/%|+1.1|+1.5|+1.1|+1.2 |+1.0 |+1.3| +1.1 |+1.4 |+6.7|
>
> * $\alpha=1$, on Cora dataset
>
> |$\beta$|0.0001|0.001|0.01| 0.1 |1.0 |10|100|1000|10000|
> |---|---| ---|---| ---| ---|---|---|---|---|
> |AM/%|92.4|92.6|92.5| 93.0 |93.4 | 93.3| 93.0| 93.4|83.3|
> |FM/%|+1.4|+1.5|+1.6|+1.4 |+1.4 |+0.9| +1.4 |+1.4 |+6.9|
>
> * $\beta=1$, on Citeseer dataset
>
>
> |$\alpha$|0.0001|0.001|0.01|0.1 |1.0 |10|100|1000|10000|
> |---|---| ---|---| ---| ---|---|---|---|---|
> |AM/%|79.0|79.7|80.0| 80.0 |80.2 |79.9 | 79.8| 80.4|75.5|
> |FM/%|+1.1|+0.5|+0.9|+0.7 |+0.8 |+0.6| +0.6| +1.0|+1.9|
>
> * $\alpha=1$, on Citeseer dataset
>
>
> |$\beta$|0.0001|0.001|0.01|0.1 |1.0 |10|100|1000|10000|
> |---|---| ---|---| ---| ---|---|---|---|---|
> |AM/%|79.0|80.0|79.8| 79.4 |80.6 |79.1 | 79.2| 80.0|75.4|
> |FM/%|+1.1|+1.1|+0.6|+1.2 |+0.7 |+1.1| +0.3| +0.8|+1.7|
>
> From these four tables we can observe that the performance is relatively robust when $\alpha$ and $\beta$ vary from 0.0001 to 1000.
>
> Through monitoring the training process, this may because that the $\mathcal{L}\_{div}$ and $\mathcal{L}\_{dis}$ decrease faster than $\mathcal{L}\_{cls}$, thus the total loss is not very sensitive to the coefficients on them.
>
> All the experiments provided above can be reproduced with our submitted code.
>
>
> Q5. Syntax errors.
>
> A5. Thanks for pointing out these errors, we will carefully correct them.

---

### Official Review · Reviewer_E15x · 2021-07-19

**Rating:** 5
**Confidence:** 5

**Summary:**

The paper addresses the continuous representation learning problem for graph data. The main goal is to learn emerging novel node categories, and simultaneously maintain the learned knowledge over existing graphs. In the proposed solution called Hierarchical Prototype Networks (HPNs), several levels of general knowledge are extracted, in the form of atomic, node and class-level prototypes. A theoretical analysis for the memory consumption and continual learning capability is presented. Empirical study is conducted  on eight different public datasets, including large-scale datasets with millions-scale nodes.

**Limitations And Societal Impact:**

Fine with me.

**Main Review:**

Strong points

- The paper addresses an important, well motivated problem in graph representation learning with practical significance.

- The presentation is largely clearly.

- Experiments are fairly comprehensive, including results on large-scale graphs.

Weak points

- There is some similarity to few-shot learning on graphs [a,b], although they mostly focus on novel class only without considering how the performance of existing classes are maintained. Nevertheless, discussion and comparison with few-shot methods should be included, to highlight the advantage of HPN over few-shot methods.

[a] Meta-GNN: On Few-shot Node Classification in Graph Meta-learning. CIKM19

[b] Relative and Absolute Location Embedding for Few-Shot Node Classification on Graph. AAAI21.

- Hierarchical graph representation learning has also been explored [c,d], which is more flexible/self-adaptive than the 3-level prototypes in this paper.

[c] Hierarchical graph representation learning with differentiable pooling. In NIPS 2018

[d] Graph Few-shot Learning via Knowledge Transfer. AAAI20.

- The model involves a number of hyperparameters related to the thresholds/number of candidates, including l_a, l_r, l'_a, l'_r, t_A, t_N, t_C. This would make model tuning slow and impractical. It is also possible that, as novel classes emerge, these hyperparameters may not to change as well for optimal performance.

- Message passing GNN is considered SOTA on graph data. The proposed solution derives representations from multi-hop neighbors, lacking the recursively defined message passing mechanism. A discussion on this is needed to demonstrate the design advantages and disadvantages of HPN.

-- update after rebuttal --

Based on the rebuttal, i have increased my rating slightly.

**Time Spent Reviewing:**

3

---

> ### Author Response · Authors · 2021-08-10
> **Response to Reviewer E15x**
>
> Thanks for raising the question about the comparison between our work and other related works. We provide below detailed explanations and discussions to clarify it.
>
> Q1. Nevertheless, discussion and comparison with few-shot methods should be included, to highlight the advantage of HPNs over few-shot methods.
>
> A1: Thanks for raising this concern. We will cite and discuss [a,b] in the updated version.
>
> Overall, as mentioned by the reviewer, few-shot learning models and HPNs are designed for different objectives. Few-shot learning models usually train a meta-learner over task 1 to n and aim to generalize to a new task (i.e., (n+1)-th task) well, while HPNs aim to learn a sequence of tasks from 1 to n such that the model's performance on previous tasks (from task 1 to n) (catastrophic forgetting) can be maintained (minimized). Specifically,
> * First, the meta-training phase of few-shot learning requires the data from multiple tasks to be present concurrently, so that the model can find parameters simultaneously suitable for different tasks. While under the continual learning setting, the tasks come in a sequential manner. The model can only access data from the current task, and is not allowed to learn multiple tasks concurrently. Therefore, the training setting of few-shot models is incompatible with the continual learning setting.
> * Second, the evaluation is different. During the meta-testing phase of the few-shot learning setting, before being tested on a task, the model is firstly fine\-tuned with data from the task (support set). In contrast, the testing phase of continual learning setting is to test the model on previous seen tasks, and no data from the test tasks would be available for the model to adapt.
>
> Given these differences, the multi-task setting of few shot learning models will be 'locked' under the continual learning setting and degenerate to pure GNN models. More concretely, [a] adopts the MAML as a general framework to train GNNs for few-shot learning. The models can be arbitrary pure GNNs without need of any special design, and the few-shot learning ability is achieved by the good parameter initialization obtained by MAML training strategy. Therefore, applying the model of [a] to a continual learning setting is equivalent to applying pure GNNs, of which the results have been shown to be inferior to HPNs in Table 1 (page 7) of our paper. Moreover, the design of model in [b] for few-shot learning is based on aggregating information from the support set with labels during testing, which is not available in continual learning testing. Therefore, without the labeled support set, the model of [b] under continual learning will also degenerate to pure GNN models without any strategy for handling different tasks.
>
>
> Q2. Hierarchical graph representation learning has also been explored [c,d], which is more flexible/self-adaptive than the 3-level prototypes in this paper.
>
> A2: Due to the different targets, the design and function of our hierarchical structure is significantly different from those ones in [c] and [d]. Our hierarchical prototypes are designed for node representations to facilitate the continual learning objective, while [c] and [d] aim at high level representations for the whole graphs or groups of nodes.
>
> More specifically, in our hierarchical prototypes, the atom-level (the lowest level) prototypes contains the decomposed node representations and is specially designed (the most important part) for the continual learning objective (Section 2.3 in paper, line 133\~189, page 4). Because these atom-level prototypes encode general characteristics shared among different types of nodes, when new categories of nodes and edges (new tasks) come in, the proposed HPNs can adaptively select, compose, and store representative embeddings with three levels of prototypes (Page 2, line 72\~73).Instead, [c] and [d] do not consider continual learning and their model does not conduct node decompositions. They mainly focus on generating high level representations for the entire graph or group of nodes. For example, the DiffPool [c] generates graph level representation for graph classification. [d] generates both class level representations representing the nodes within the same class and graph level representations for the entire graphs.
>
> Overall, although [c] and [d] share some similar ideas with HPNs (with respect to hierarchical structure), they cannot be directly applied for tasks in this paper due to different targets and design logics. Moreover, our theoretical results (Section 2.5 in paper, line 206~237, page 6) and the experimental results (Section 3 in paper, line 238, page 6) demonstrated the effectiveness of HPNs for the continual learning task.
>
>
> Q3. The model involves a number of hyperparameters related to the thresholds/number of candidates, including $l_a, l_r, l'_a, l'_r, t_A, t_N, t_C$. This would make model tuning slow and impractical.
>
> A3: As shown in the experiments on multiple datasets (Section 3.6 in paper, Figure 2, Section 3.4 in Appendix, Figure 3), the model is not very sensitive to hyperparameters ($t_A$, $t_N$, $t_C$, $l_a$, $l_r$, etc.). Also, the same hyperparameters were used across different datasets for comparison experiments with the baselines, and the results demonstrated the robustness of the proposed HPNs (Section 3.2 in paper, line 261, page 7; Section 2.2 in Appendix, line 254\~273, page 7 ).
>
> Specifically, the sensitivity on the thresholds $t_A$, $t_N$, and $t_C$ are shown in Section 3.6 in paper  (line 331\~336, page 9) and Section 3.4 in Appendix (line 321\~323, page 10). The results demonstrated that the model is robust against different thresholds, and feasible choices are consistent for different datasets. Experiments on different choices of the number of AFEs and the embedding dimensions ($l_a$, $l_r$, etc.) are provided in Section 3.6 in paper (line 320\~330, page 9). The results also showed a wide range of feasible choices for these hyperparameters.
>
>
> Q4. It is also possible that, as novel classes emerge, these hyperparameters may not change as well for optimal performance.
>
> A4: The new tasks are accommodated by the trainable parameters of the model, like the AFEs and the prototypes, and do not require the hyperparameters to be changed accordingly. Similar to other continual learning works [4,5,6], as shown on 8 datasets in our experiments, the hyperparameters are fixed during the whole training process with up to tens of new classes keeping emerging, our model performs consistently well on all datasets.
>
>
> Q5. The proposed solution derives representations from multi-hop neighbors, lacking the recursively defined message passing mechanism. A discussion on this is needed to demonstrate the design advantages and disadvantages of HPN.
>
> A5: The message passing mechanism is a popular approach to aggregate information from multi-hop neighbors. However, with the recursively defined message passing process, when generating a representation for a node $v$, the information from distant nodes has to pass intermediate nodes before reaching node $v$, which mixes the information from different neighbors. In our paper, as we aim to distill prototypes to represent the basic relation types shared among different tasks, the pairwise relationship between node $v$ and each of the neighbors in the multi-hop neighborhood has to be captured separately. Therefore, the message passing mechanism is not feasible for our model, as it mixes the information from different neighbors.
>
> Actually, as long as the multi-hop neighbor information can be captured, directly deriving the information is as good as or even better than the recursively defined message passing in practice. Several recent works on designing direct neighborhood aggregation without the recursive manner [1,2,3] have demonstrated this point.
>
> Moreover, the GNNs with message passing mechanisms are already included in our baselines (GCN, GAT, and GIN in Section 3.3 of the paper, line 266\~289, page 7), and the results demonstrated the effectiveness of our proposed HPNs.
>
> Finally, from Table 3 and 4 of the Appendix (line 298, page 9), we see that even on individual tasks without considering the continual learning objective, our model is slightly better than the GAT and GCN models (which leverage message passing mechanism) on most tasks. This demonstrates the effectiveness of the strategy of leveraging multi-hop neighbors in HPNs.
>
>
> [1] Wu, F., Souza, A., Zhang, T., Fifty, C., Yu, T. and Weinberger, K., 2019, May. Simplifying graph convolutional networks. In International conference on machine learning (pp. 6861-6871). PMLR.
>
> [2] Zhang, X., Liu, H., Li, Q. and Wu, X.M., 2019. Attributed graph clustering via adaptive graph convolution. arXiv preprint arXiv:1906.01210.
>
> [3] Zhu, H. and Koniusz, P., 2020, September. Simple spectral graph convolution. In International Conference on Learning Representations.
>
> [4] Zhou, Fan, and Chengtai Cao. "Overcoming Catastrophic Forgetting in Graph Neural Networks with Experience Replay." Proceedings of the AAAI Conference on Artificial Intelligence. Vol. 35. No. 5. 2021.
>
> [5] Liu, H., Yang, Y. and Wang, X., 2020. Overcoming catastrophic forgetting in graph neural networks. arXiv preprint arXiv:2012.06002.
>
> [6] Parisi, G.I., Kemker, R., Part, J.L., Kanan, C. and Wermter, S., 2019. Continual lifelong learning with neural networks: A review. Neural Networks, 113, pp.54-71.

---

> ### Author Response · Authors · 2021-08-20
> **We are happy to address any further concerns**
>
> Thanks a lot for raising the questions on the difference between our work and other works. We have provided detailed responses to these questions and hope you are satisfied with our answers.  If there are any further concerns, please feel free to let us know and we are more than happy to address them.

---

> ### Author Response · Authors · 2021-08-25
> **Remaining concerns/questions？**
>
> We thank the reviewer for raising the score. We would appreciate it if you could share the remaining concerns/questions given our previous responses. We will be more than happy to clarify them.

---

### Official Review · Reviewer_9787 · 2021-07-22

**Rating:** 5
**Confidence:** 3

**Summary:**

This paper addresses graph representation learning in a particular online/continual/streaming setting in which subgraphs are sequentially recieved which may contain nodes of categories/classes that haven't been seen before, while performance on the previously seen categories/classes must be maintained.  The approach involves extracting "atomic" features based on node features and local network structure, then constructing and maintaining prototype embeddings at various levels of abstraction, in order to produce a final embedding to be used by a node classifier. In each iteration the method decides whether to introduce new prototypes based on matching to previous prototypes. The authors prove a bound on the number of prototypes (hence, memory requirements) and show that the method will avoid catastrophic forgetting. Experimental results versus various baselines are positive.

**Ethical Concerns:**

No ethical concerns were identified.


**Limitations And Societal Impact:**

Graph representation learning can be subject to algorithmic bias/fairness issues.  It would be worth mentioning how this might be addressed in future work.


**Main Review:**


Originality: While graph representation learning has been heavily studied, including substantial work on temporal/dynamic networks, and continual/online learning has also received significant attention, the particular setting addressed in the paper is an interesting special case of the problem.  The proposed method has not been considered before, but it is not clear that broader insights arise from, or underpin, its development.

Quality: Although intuitively appealing, the proposed model design is somewhat heuristic in nature.  It is unclear what fundamental principles motivate the use of prototypes or the various algorithmic choices which were made.  At least, the theoretical results in Section 2.5 show that the proposed method avoids certain pitfalls, but they do not otherwise guarantee how close to optimal the predictions are (e.g., in terms of regret).  The experimental results are a strength of the work, including good results in a systematic cross-validation-with-hyperparameter-tuning experiment compared to a number of baselines on multiple datasets, an ablation study, memory consumption, and t-SNE visualizations.

Clarity: The paper can be understood, with some effort.  There are relatively few typos or grammatical issues.  Figure 1, explaining the method, is helpful in understanding the approach.

Significance: The strong experimental results suggest that the proposed method is an improvement over the current state of the art at this particular representation learning problem, which may have practical importance.  However, the heuristic nature of the approach does not suggest that the ideas here will have influence beyond the narrow task, or will stand the test of time.

------

Thanks to the authors for their detailed and well argued rebuttal. The response was helpful for understanding how the authors perceive the points I've raised.   I have no problem with the hierarchical prototype approach proposed here, but I don't really see an overarching objective or motivating principle that spans both the matching prototypes phase and the learning phase, each of which seems to have unrelated objectives.

**Time Spent Reviewing:**

3 hours

---

> ### Author Response · Authors · 2021-08-10
> **Response to Reviewer 9787**
>
> We sincerely thank the reviewer for the constructive comments, we provide below detailed responses to clarify these questions.
>
> Q1: It is unclear what fundamental principles motivate the use of prototypes or the various algorithmic choices which were made.
>
> A1:  Thanks for the question. In this work, we aim to address the problem of continual graph representation learning in which new categories of nodes and their associated edges are continuously emerging. The key idea is to incorporate the feature and topological information of new nodes in a continuous and effective manner such that performance over existing nodes is uninterrupted (line 4\~6). In this case, prototype based networks are a natural choice since prototypes can be adjusted independently (when necessary) which is desired for the objective of continual learning. Specifically, when each new task comes in, we only need to adjust the relevant prototypes without modifying the whole model to avoid catastrophic forgetting. This benefit on continual learning is also justified by theoretical analysis (Theorem 2 of Section 2.5 in paper, line 224, page 6) and empirical studies (Section 3.3 in paper, line 266\~289, page 7; Section 3.5 in paper, line 301\~318, page 9).
>
> Meanwhile, the usage of hierarchical prototypes is inspired by the fact that humans learn to recognize objects by forming prototypes in the brain [1,7,8] (the three mainstream continual learning approaches are also inspired by neuroscience findings [2,3,4]). Humans formulate real world objects into hierarchical representations and can recognize a tremendous number of objects without forgetting. The hierarchical representations include basic properties shared by different objects like color, shape, hardness, etc., a middle-level representation indicating the whole object, e.g. apple, pear, and a high-level concept like fruit. Therefore, similar to this idea, we decompose graph nodes in a hierarchical manner, and design the atomic-level (e.g., node attributes such as gender, nationality, hobby, etc.), node-level, and class-level prototypes for node representations accordingly.
>
> Moreover, our prototype based framework is also designed to avoid certain drawbacks of the mainstream continual learning approaches (Section 4.1 in Appendix, line 342\~387, page 11) while keeping their advantages simultaneously.
> * First, compared with the memory replay methods, both frameworks can store the information from the previous tasks. But the memory replay based methods have to retrain the model with data exemplars from previous tasks each time when a new task comes in, which causes extra computational burden. On the contrary, our prototype based learning process integrates the information of the previous tasks into the inference process and does not have to be retrained.
> * Second, both the parameter isolation based approaches and our model can allocate different parameters for different tasks. But the parameter isolation based approaches have to isolate certain parameters for each task, causing the model memory unbounded. In contrast, our model shares the prototypes among all tasks, thus the parameters are highly reusable across tasks and the model is much more memory efficient. Also, our model has a theoretical memory consumption upper bound (Theorem 1 in Section 2.5 of the paper, line 206\~223, page 6), which is also experimentally validated (Section 3.7 in paper, line 337\~346, page 9; Section 3.5 in Appendix, line 324\~331, page 10).
>
> Finally, the choices of other algorithmic designs are also explained and experimentally validated in the paper, like the AFEs (Introduction of paper, line 66\~72, page 2; Section 2.2 in paper, line 112\~132, page 3; Section 3.6 in paper, line 325\~330, page 9), different loss terms (Section 2.4 in paper, line 190\~205, page 5; Section 3.4 in paper, line 295\~300, page 8), etc.
>
> In summary, our proposed prototype based framework is motivated by the problem setup, neuroscience findings, and theoretical justifications. The motivations of other algorithmic designs are also explained and backed up by experiments. In particular, our work is one of the few works with theoretical justification on the continual learning capability (the only one on continual graph learning), and may propose a promising direction to theoretically analyze the continual graph learning models.
>
>
> Q2. The theoretical results in Section 2.5 show that the proposed method avoids certain pitfalls, but they do not otherwise guarantee how close to optimal the predictions are (e.g., in terms of regret).
>
> A2:  Thanks for raising this concern. For continual learning, optimality is concerned with two aspects: the minimization of forgetting across tasks and the maximization of performance on individual tasks. Our Theorem 2 provides insights on bounding the forgetting (Section 2.5 in paper, line 224\~237, page 6), which is the primal objective of continual learning.
>
> As we have no constraints (like the regularization based continual learning approaches) on the model when learning new tasks, our designs for continual learning will not affect the model's performance on individual tasks. Therefore, the optimality of the model's performance on individual tasks is orthogonal to the continual learning objective and thus not analyzed in this work.
>
> Analyzing the proposed HPNs from the perspective of regret is an interesting direction, which can increase the consistency of our theoretical analysis to the broader machine learning community. This can be accomplished by extending our Theorem 2. We will conduct corresponding analysis to further improve our work in the future.
>
>
> Q3.  However, the heuristic nature of the approach does not suggest that the ideas here will have influence beyond the narrow task, or will stand the test of time.
>
> A3: Thanks for the question. The setting adopted in our model is actually related to many real world scenarios and has started to attract increasingly more attention recently [9,10,11]. For example, with the emergence of new research domain, the citation network is constantly expanded with new subgraphs with novel node classes (like the OGB-Arxiv dataset in our experiments); with the arrival of new types of products, the co-purchasing network is also expanded with new subgraphs with new node classes (like the OGB-Products dataset containing the Amazon co-purchasing network in our experiments). In these scenarios, a model has to not only learn to classify new classes of nodes, but also retain the ability to recognize the old classes, as the old classes of nodes may still emerge (Introduction of paper, line 30\~36, page 1).
>
> Besides the practical importance, our theoretical framework may also provide a promising direction for future works on theoretically analyzing continual graph learning or general continual learning models.
>
> Overall, we believe our work can make a good contribution to the community and may trigger many following works in this direction.
>
>
> Q4. Algorithmic bias/fairness issues.
>
> A4: Thanks for this constructive suggestion. The algorithmic bias/fairness are important issues to consider and we will add more discussions on them.
>
> Currently, for the node classification task in our paper, possible algorithmic bias is caused by the imbalanced datasets. To deal with this, we reordered the classes according to the number of examples of each class, and group the classes of similar sizes into the same task to minimize the class imbalance within each task (Section 2.1 in Appendix, line 182~243, page 6). We will also add more discussions on the algorithmic bias that may emerge in other tasks [4,5] based on our representation learning framework.
>
>
> [1] Bowman, C.R., Iwashita, T. and Zeithamova, D., 2020. Tracking prototype and exemplar representations in the brain across learning. Elife, 9, p.e59360.
>
> [2] Parisi, G.I., Kemker, R., Part, J.L., Kanan, C. and Wermter, S., 2019. Continual lifelong learning with neural networks: A review. Neural Networks, 113, pp.54-71.
>
> [3] Kirkpatrick, J., Pascanu, R., Rabinowitz, N., Veness, J., Desjardins, G., Rusu, A.A., Milan, K., Quan, J., Ramalho, T., Grabska-Barwinska, A. and Hassabis, D., 2017. Overcoming catastrophic forgetting in neural networks. Proceedings of the national academy of sciences, 114(13), pp.3521-3526.
>
> [4] Zenke, F., Poole, B. and Ganguli, S., 2017, July. Continual learning through synaptic intelligence. In International Conference on Machine Learning (pp. 3987-3995). PMLR.
>
> [5] Dai, E. and Wang, S., 2021, March. Say no to the discrimination: Learning fair graph neural networks with limited sensitive attribute information. In Proceedings of the 14th ACM International Conference on Web Search and Data Mining (pp. 680-688).
>
> [6] Li, P., Wang, Y., Zhao, H., Hong, P. and Liu, H., 2020, September. On dyadic fairness: Exploring and mitigating bias in graph connections. In International Conference on Learning Representations.
>
> [7] Dandan, T., Haixue, Z., Wenfu, L., Wenjing, Y., Jiang, Q. and Qinglin, Z., 2013. Brain activity in using heuristic prototype to solve insightful problems. Behavioural brain research, 253, pp.139-144.
>
> [8] Zeithamova, D., Maddox, W.T. and Schnyer, D.M., 2008. Dissociable prototype learning systems: evidence from brain imaging and behavior. Journal of Neuroscience, 28(49), pp.13194-13201.
>
> [9] Zhou, Fan, and Chengtai Cao. "Overcoming Catastrophic Forgetting in Graph Neural Networks with Experience Replay." Proceedings of the AAAI Conference on Artificial Intelligence. Vol. 35. No. 5. 2021.
>
> [10] Liu, H., Yang, Y. and Wang, X., 2020. Overcoming catastrophic forgetting in graph neural networks. arXiv preprint arXiv:2012.06002.
>
> [11] Kou, X., Lin, Y., Liu, S., Li, P., Zhou, J. and Zhang, Y., 2020. Disentangle-based Continual Graph Representation Learning. arXiv preprint arXiv:2010.02565.

---

> ### Author Response · Authors · 2021-08-20
> **We are happy to address any further concerns**
>
> We sincerely appreciate the reviewer's effort and constructive comments. We have provided detailed responses to clarify all questions. If you have any further concerns, please feel free to let us know and we are more than happy to answer them.

---

> ### Author Response · Authors · 2021-08-25
> **Responses to further concerns**
>
> We sincerely thank the reviewer for raising the concern on how the matching prototype phase and learning phase are related to the overarching objective.
>
> Since HPNs are developed for continual graph representation learning, our overarching aim is to avoid forgetting by only optimizing the prototypes relevant to the current task without influencing the prototypes for previous tasks. Specifically, our aim is accomplished by two consecutive procedures:
>
> 1. detecting the relevant prototypes.
>
> 2. optimize the relevant prototypes.
>
> The prototype matching phase and the learning phase are designed for these two procedures, respectively. The prototype matching phase serves to find the prototypes relevant to the data in the current task, and then the matched prototypes are optimized by the learning phase (Introduction, line 59\~90, page 2). Our Theorem 2 in Section 2.5 also explores the collaboration between the matching prototype phase and the learning phase, i.e. how the matching thresholds $t_A, t_N, t_C$ adjust the amount of forgetting caused by the learning phase, which is also explored in the experiments  (line 331\~ 336, page 9).
>
> All relevant explanations can be found in the paper, e.g. the overarching objective with explanations on the matching prototypes and learning phase is in Introduction (line 59\~90, page 2) and Section 2 (line 96\~99, page 2); the technical details on how the prototype matching phase and learning phase collaborate to avoid forgetting are in Section 2.3 (line 142\~168, page 4; line 176\~178, page 5); how the thresholds used for prototype matching affect the continual learning performance is both theoretically analyzed in Section 2.5 (line 224\~232, page 6) and experimentally explored in Section 3.6 (line 331\~ 336, page 9).
>
> In summary, our overarching objective is to avoid forgetting under the continual learning setting, and the two phases mentioned by the reviewer all serve this target. Their mechanisms are also backed up with theoretical and experimental results.
>
> Note that our design logic to avoid forgetting are also recognized by other reviewers (in the summary of Reviewer eQnQ and eRkd)
>
> Thanks again for the constructive comments. We will emphasize and make it more clear in the updated version. If you have any further concerns/questions, please feel free to let us know and we are more than happy to answer them.

---

### Author Response · Authors · 2021-08-31
**Summary of the reviews and our responses**

Dear Reviewers and ACs/SACs/PCs,

We authors appreciate your time, efforts, and constructive suggestions in the reviewing process. To further facilitate the discussion, we summarize below the major strengths and remaining concerns raised by the reviewers and our responses.

$\textbf{Major Strengths:} $

1. All four reviewers agree that our work addresses a $\textit{challenging/well-motivated/important/interesting}$ graph representation learning problem with practical $\textit{significance/importance}$ and $\textit{strong/comprehensive/thorough}$ experimental results on wide range of heterogenous datasets including large scale ones.

2. Most reviewers (Reviewer eQnQ, E15x, and 9787) found our paper $\textit{overall clear and easy to follow/largely clear/with relatively few typos or grammatical issues}$.

$\textbf{Remaining concerns:} $

After submitting our responses, Reviewer 9787 raised an additional concern on the overarching objective of our designs. We responded with detailed explanations on what our overarching aim is and how each component is designed to serve this aim. We also referred the reviewer to related sections in our paper for detailed explanations.

$\textbf{Original major concerns and responses:}$

Reviewer 9787:

Q1. What are the fundamental principles motivating the designs in our model?

We first clarified our fundamental aim to address the continual graph representation learning problem. Then we explained how the designs of our model serve this purpose. Besides, we also strengthened the theoretical and experimental foundations of these designs as well as the neural science supports. Finally, we explained how these designs were formulated to also avoid the limitations of existing mainstream continual learning approaches.

Q2. How close to optimal are the predictions?

We explained how our theoretical analysis reflects the optimality, as well as how to enrich our theoretical analysis with the suggestions from the reviewer.

Reviewer E15x:

Q1. Highlighting the advantages of HPNs over few-shot graph learning works is needed.

We explained the difference between our setting (continual learning) and few-shot learning setting, showing that the models for these two settings are not interchangeable and infeasible to be directly compared for advantages.

Q2. Some hierarchical graph representation learning works have more flexible hierarchical structures

We explained the different aims and design logics of our model and those hierarchical graph representations learning works, which showed that those models cannot be directly applied to our continual learning task.

Q3. Why not use recursive message passing based MPNNs and what are the advantages of our model?

We explained the advantages of our approach and why the recursive message passing was not suitable for our model. Besides, MPNN models were already included as baselines and the experiments demonstrated the effectiveness of our approach.

Q4. Tuning the hyperparameters may be difficult.

In our paper, we designed experiments on multiple datasets to show that our model is not very sensitive to hyperparameters and does not need much tuning. Besides, the same hyperparameters were used across different datasets (without tuning for each dataset) for comparisons with the baselines, and the results demonstrated the robustness of our model.

Reviewer eQnQ is mainly concerned with some technical details, as listed below:

Q1. Whether parameter saving claim holds for datasets without node features?

On datasets without node features, using HPNs with small embedding dimensions can ensure the parameter saving, of which the performance was shown in our experiments.

Q2. How the hard selection is made differential?

The hard selection process is non-differentiable and is used as a preprocessing step. We made the training differentiable by using the distance loss.

Q3. Why not add coefficients to balance different loss terms?

The model is not sensitive to the coefficients on the loss terms, as shown in additional experiments. For simplicity, we omitted the coefficients.

Reviewer eRkd:

Q1. Explanations needed on some details, including the technical ones like the idea behind neighbor subsampling, and others like the location of the proof.

We provided explanations on each of the concerns with references to the corresponding parts in the paper.

Q2. Concern on the difference between performance of a baseline in our paper and in the original paper

We explained the different metrics used in our paper and in the original paper with concrete examples from the experimental results. We also referred to milestone continual learning works to show that the metric in our paper is adopted by nearly all continual learning works.

---

### Decision · Program_Chairs · 2021-09-27

**Decision:**

Reject

**Comment:**

The paper addresses graph representation learning when new categories of nodes emerge continually along with their associated edges. The idea is to learn a Hierarchical Prototype Networks (HPNs) which extract prototypes at different levels (atom, node, class) to represent the continually growing graphs. This allows updates to focus on relevant prototypes and atomic feature extractors. Analysis on memory consumption is given, along with continual learning capability.

Although the proposed approach is a bit ad hoc, it has interesting ideas and the empirical performance looks fine.  Three major drawbacks are identified by the reviewers, and I agree with them. Firstly, the presentation can be improved to clarify the motivation, and to provide an overarching objective or motivating principle that spans both the matching prototype phase and the learning phase (quote Reviewer 9787).  Section 2.2 and 2.3 read very dry, and Reviewer eRkd provided very detailed comments.  Secondly a detailed discussion on the large number of hyperparameters is needed.  Finally, although [c, d] raised by Reviewer E15x do not directly work on continual learning, the few-shot nature makes such adaptation quite straightforward.  A detailed comparison is needed on such extensions.

All reviewers except the most positive one (eQnQ) participated in discussions that are not visible to the author.

Minor:
Equation 4 needs to reduce to a scalar
Algorithm has duplicated lines 4 and 5